# Fluorinated Polyimide-Film Based Temperature and Humidity Sensor Utilizing Fiber Bragg Grating

**DOI:** 10.3390/s20195469

**Published:** 2020-09-24

**Authors:** Xiuxiu Xu, Mingming Luo, Jianfei Liu, Nannan Luan

**Affiliations:** 1School of Electronic and Information Engineering, Hebei University of Technology, Tianjin 300401, China; 201921902022@stu.hebut.edu.cn (X.X.); jfliu@hebut.edu.cn (J.L.); luan@hebut.edu.cn (N.L.); 2Tianjin Key Laboratory of Electronic Materials and Devices, Tianjin 300401, China; 3Hebei Key Laboratory of Advanced Laser Technology and Equipment, Tianjin 300401, China

**Keywords:** fluorinated polyimide film, humidity hysteresis, fiber Bragg grating

## Abstract

We propose and demonstrate a temperature and humidity sensor based on a fluorinated polyimide film and fiber Bragg grating. Moisture-induced film expansion or contraction causes an extra strain, which is transferred to the fiber Bragg grating and leads to a humidity-dependent wavelength shift. The hydrophobic fluoride doping in the polyimide film helps to restrain its humidity hysteresis and provides a short moisture breathing time less than 2 min. Additionally, another cascaded fiber Bragg grating is used to exclude its thermal crosstalk, with a temperature accuracy of ±0.5 °C. Experimental monitoring over 9000 min revealed a considerable humidity accuracy better than ±3% relative humidity, due to the sensitized separate film-grating structure. The passive and electromagnetic immune sensor proved itself in field tests and could have sensing applications in the electro-sensitive storage of fuel, explosives, and chemicals.

## 1. Introduction

Humidity monitoring and regulation are of great significance in pharmacy, semiconductors, and costly facility maintenance and explosive storage [1,2,3]. Relative humidity, defined as the ratio of vapor content in the air, describes the present moisture content compared with what the air can hold at most [4]. Mechanical psychrometers and dew-point hygrometers are limited in humidity sensing due to their low accuracy and complexity [1]. Chemical methods mainly depend on the hygroscopic reaction, which has been found to be irreversible and nonreusable [5]. Thus, capacitance and resistance are selected as measurable signs of humidity change, which are accurate, fast-responding, and small-scaled [3]. However, for safety reasons, active electronic humidity sensors prove unsuitable for electro-sensitive applications such as explosives, chemicals and fuel storage [1,2,3,4].

Optical fiber sensing, as a passive and electromagnetism-immune method, provides a safe and reliable approach to temperature and humidity monitoring for such electro-sensitive storage [3,6]. However, different mechanisms, including spectral absorption, evanescent sensing, and light scattering, show poor performance in terms of flexibility and multiplexing [7,8,9,10,11,12,13]. Additionally, non-uniform sensor structures, such as the Fabry-Perot cavity [7], fiber taper [8], micro-nano fiber [9], side-polished fiber [10], and U-bent fiber [11], are fragile and unfit for mass production [12,13,14,15,16]. Moreover, the combination of the functional materials and fiber sensors shows non-uniformity and uncertainty as well, especially when it comes to graphene oxide [17], metal oxide [18], UV gels [19], and nanoparticles [20]. For practical applications, fiber Bragg grating (FBG) combined with polymer provides a better way due to its long service and uniformity in mass production. The humidity-dependent polyimide film was first investigated by Kronenberg, revealing its potential application in relative humidity (RH) sensing [21]. Later, he demonstrated a humidity sensor with polyimide coating at the grating by measuring the moisture-induced expansion [22]. T. Yeo and K. Grattan analyzed the strain transfer between the polyimide coating and the fiber Bragg grating, and explained the relationship between humidity and wavelength shift in detail [23]. The linear sensitivity was estimated to be +4.5 pm/RH with an uncertainty of ±4% RH, while the polyimide coating resulted in humidity hysteresis and a response time over 40 min. Recently, similar attempts have been made for a higher sensitivity and a shorter breathing time, which are hard to achieve simultaneously with an ordinary polyimide film in a conventional structure. The reason lies in the tough hydrogen bonding in polyimide with a porous surface, which causes humidity hysteresis and even moisture agglomeration [24,25]. Moreover, the moisture exchange and humidity sensitivity are restricted with each other depending on the tradeoff between film thickness and surface, where it is difficult to balance the response time and accuracy. According to the aforementioned analysis, a humidity sensor with high performance still remains to be realized both in chemical modification and structural design.

In this paper, a fluorinated polyimide-film-based temperature and humidity sensor utilizing fiber Bragg grating is proposed and demonstrated. The sensor was constructed with an FBG and a 20 μm-thick fluorinated polyimide film by direct manual gluing. Moisture-induced film expansion or contraction causes an extra strain, which is transferred to the FBG and leads to a humidity-dependent wavelength shift. Compared with the coating method, the separate film-grating structure provides a large surface area for moisture exchange, an improved dynamic range, and considerable sensitivity. The –CF_3_ modification in the monomer of the polyimide reduces its moisture capacity, restrains its humidity hysteresis, and provides a short moisture breathing time less than 2 min. Another cascaded FBG temperature sensor is used to exclude its thermal crosstalk, with a temperature accuracy of ±0.5 °C. Experimental monitoring over 9000 min revealed a relative humidity accuracy better than ±3% RH, while field testing for tobacco storage proved its stability and practicability as designed. The passive and electromagnetism-immune sensor with an accurate and fast response can be applied for the electro-sensitive storage of explosives, chemicals, and fuels.

## 2. Materials and Methods

### 2.1. Functional Material

For common polyimide films, the vapor in the air is easily captured by the hydrophilic hydrogen bonds and porous surface. The strong absorption causes a slow moisture release and may even lead to humidity hysteresis and agglomeration. Thus, chemical modification is necessary to improve the surface properties of the polyimide film. The fluorinated polyimide film was synthesized and provided by the group of Pro. L. Fan, Institute of Chemistry Chinese Academy of Sciences, and is commercially available with excellent hysteresis resistance [26]. As can be seen from the chemical formula and the molecular structure of the fluorinated polyimide shown in Figure 1a, the trifluoromethyl modification replaces the ether bond in the original aromatic monomer with a molecular weight of 630. In the detailed straight-chain polycondensation, the fluorine modification improves the hydrophobicity of the film [26,27]. With the increase in fluorine content from 15.32% to 30.16% in the monomer, the transparent film becomes an applicable candidate for a fast-response humidity sensor. Compared with a non-fluorinated film synthesized with a similar monomer with a molecular weight of 496 such as the dark one shown in Figure 1b, the trifluoromethyl acts in a moisture-proof manner and restrains its capacity from 5% to 1%, which is quantified by weight gain at saturated humidity. Thus, with such a hydrophobic modification, a small humidity hysteresis and a short breathing time can be expected when using a fluorinated polyimide film.

As the moisture capacity is significantly reduced in a fluorinated polyimide film, the sensitivity to humidity declines sharply along with the insufficient expansion. The conventional coating method is no longer suitable for practical use, with which it is hard to achieve a fast response, a low hysteresis, and considerable sensitivity simultaneously.

### 2.2. Sensor Structure

Thus, a separate film-grating structure was specifically designed to maintain the dynamic range and the humidity sensitivity, as a result of the large cross-sectional ratio between the functional film and fiber grating. As shown in Figure 2a, the sensing unit was fabricated with a bare FBG of 125 μm in diameter and a film of 20 μm in thickness. The FBG and the film were assembled with an aluminous convertor and clamped between two brackets of a hollow frame by direct manual gluing. The unit was pre-stretched at 2000 με to ensure an approximately linear spectral response.

Once the humidity changes, the film self-rebalances by exchanging moisture with that in the air. In this way, the expansion or contraction leads to an axial strain change transferred to the FBG. Finally, the humidity variation is visualized by tracing the wavelength shift, which indicates dynamic moisture absorption and release. Moreover, Figure 2b shows another cascaded FBG temperature sensor to exclude the thermal crosstalk and extract the pure relative-humidity changes from the co-effects on the wavelength shift, as circled with the red dotted line.

### 2.3. Working Principles

The humidity sensing mechanism is further discussed below. According to the equal stress in a rigid connection as in Equation (1), the strain transferred to the FBG is strictly associated with the expansion and contraction of the polyimide film,
(1)$$E_{\mathrm{pi}}\varepsilon_{\mathrm{pi}}W_{\mathrm{pi}}d_{\mathrm{pi}}=E_{G}\varepsilon_{G}\pi R_{G}^{2}$$
where *ε*_pi_ (*ε*_G_) and *E*_pi_ (*E*_G_) refer to the strain and the elastic modulus of the film (grating), respectively. *W*_pi_ and *d*_pi_ describe the transverse section of film, while *R*_G_ denotes the radius of the bare fiber. Besides, the polyimide film expansion strictly equals the contraction in the FBG, fixing an isolated length at a constant temperature in Equation (2).
(2)$$L_{\mathrm{pi}}\alpha_{\mathrm{pi}}H(1+\varepsilon_{\mathrm{pi}})+L_{\mathrm{pi}}\varepsilon_{\mathrm{pi}}+L_{G}\varepsilon_{G}=C_{0}$$

*α*_pi_ and *H* represent the hygroscopic expansion coefficient and the humidity variation, and *L*_G_ and *L*_pi_ stand for the lengths of the fiber grating and film available, respectively. After derivation calculus on Equation (2) and simplification with Equation (1), we deduce the differential sensitivity *S_H_* as the humidity-dependent wavelength shift,
(3)$$S_{H}=\frac{d\lambda_{G}}{dH}=k\frac{d\varepsilon_{G}}{dH}=-k\frac{E_{\mathrm{pi}}W_{\mathrm{pi}}d_{\mathrm{pi}}}{E_{G}\pi R_{G}^{2}}\frac{1+\varepsilon_{0}}{H+\frac{1}{\alpha_{\mathrm{pi}}}+\frac{1}{\alpha_{\mathrm{pi}}}\frac{L_{G}}{L_{\mathrm{pi}}}\frac{E_{\mathrm{pi}}W_{\mathrm{pi}}d_{\mathrm{pi}}}{E_{G}\pi R_{G}^{2}}}$$
where *k* is defined as the transfer coefficient between the axial strain and wavelength shift, and *ε*_0_ refers to the initialized strain in the FBG. The negative expression indicates a blue-shift direction with an alterable sensitivity, which can be enhanced by increasing the cross-sectional ratio and the length ratio between the film and FBG. A slight nonlinearity in the sensitivity is introduced at different humidity, which can be well corrected with a second-polynomial coefficient.

## 3. Results

The temperature and humidity sensors were calibrated with a highly accurate Humidity Generator (GEO Calibration, Model 2000), and their sensing performance was validated in humidity-mimicking enclosures created with different saturated solutions. As shown in Figure 3, the sensors were exposed in different humidity enclosures of 12% (LiCl), 33% (MgCl_2_), 60% (NaBr), 75% (NaCl), and 98% (K_2_SO_4_). The wavelengths measured at humidity of 10.4%, 34.2%, 57.6%, 76.1%, and 96.2% at 25 °C were distributed near the calibration fitting curve, similar with those at other temperatures.

As assumed in theoretical analysis, the polynomial fitting curves exhibit an approximately linear dependence between the humidity and the wavelength shift. Additionally, the slight nonlinearity can be estimated with a second coefficient as the humidity varies from 12% RH to 98% RH. Its central wavelength shift towards the blue direction from 1535.1 to 1534.6 nm agrees well with the looseness of the polyimide film and decrement in axial strain. The temperature and humidity sensitivities were obtained as +5.033 pm/°C and −6.09 pm/%RH, respectively, providing opposite spectral responses to better exclude the thermal crosstalk.

The spectral response to humidity change is compared with that of a CE314 electronic sensor, and the humidity hysteresis is discussed in Figure 4. Both the FBG sensor and the electronic sensor were exposed in the same enclosures of the LiCl (12% RH) and K_2_SO_4_ (98% RH) solutions, as well as in the lab (62% RH). The red line above indicates the humidity recorded by CE314, while the blue line represents the wavelength shift determined with the FBG humidity sensor. As the humidity changes among 12% RH, 98% RH, and 62% RH, the central wavelength of the FBG exhibits a red-shift or blue-shift in response to a decrement or increment in the humidity, respectively. Due to the fluoride doping in the polyimide film, a short breathing time less than 2 min is observed with low hysteresis. Besides, the considerable sensitivity and accuracy are preserved, as a consequence of the separate film-grating structure. Thus, the humidity sensor is experimentally proved to be feasible with considerable sensitivity, a fast response, and low humidity hysteresis.

In the experiment as shown in Figure 5, the sensors were applied for practical monitoring after aging for one month. The signals from the Amplified Spontaneous Emission (ASE) source were modulated by temperature and humidity sensors, harvested by an Optical Spectral Analyzing (OSA) module, and demodulated with pre-recorded calibration coefficients. The temperature and relative humidity visually appeared on the display and were continuously recorded backstage.

Figure 6 shows the temperature varying from 17 to 35 °C for the FBG sensors and an electronic sensor CE314, which is consistent between each. The deviations between the black, red, green, and blue lines demonstrate the temperature differences over 9000 min of monitoring. Mostly, the temperature deviations are well restricted within ±0.5 °C, while such errors beyond tolerance occasionally occur at the perturbations as circled by the dashed lines above.

Figure 7 shows the humidity variation aligned with the monitoring timeline of the temperature. The humidity is also demonstrated with black, red, green, and blue lines, which are consistent with each other as well. Basically, the humidity deviations are restricted within ±3% RH, while the errors mainly occur at temperature jumping.

Moreover, as shown in Figure 8, the practicability was verified through field tests in a tobacco warehouse, Fujian province, Southeast of China. As the tobacco standards require, the unprocessed tobacco should be stored at 20 to 30 °C, and 65% RH to 75% RH. Three FBG temperature/humidity sensors were placed in Storage 1, Storage 2, and the outdoor area, which were connected to Channel 1, Channel 2, and Channel 3 via commercial cables with an insertion loss of about 1.1 dB. The backward optical signals were harvested by a distant interrogator and demodulated to real-time temperature and humidity. During 18 h of discontinuous monitoring over 3 days, data were recorded by the FBG sensors and are expressed with solid lines, while those from the manual inspections with CE314 are depicted with scatter points. By comparison, the FBG sensors agreed well with the electronic sensor, whether in storage or outdoors. Since the circumstance in the storage was relatively stable, the temperature varied from 27.5 to 28.4 °C and the humidity changed from 68% RH to 74% RH. Due to the airflow perturbations outdoors, the temperature varied from 28.7 to 35.3 °C and the humidity changed from 46% RH to 73% RH. The deviation of T ≤ 0.3 °C and H ≤ 2.8% RH outdoors is larger than that in storage, of T ≤ 0.2 °C and H ≤ 2.2% RH, which is also consistent with actual situations.

## 4. Conclusions

We propose and demonstrate a fluorinated-film-based FBG temperature and humidity sensor. The fluoride doping resolves the humidity hysteresis and provides a short breathing time less than 2 min. The calibrated sensors show different humidity corresponding to enclosures created with saturated solutions and good stability during 9000 min of monitoring in the lab. Additionally, the field tests prove the outstanding performance and practicability of the humidity sensors, but the consistency, size, weight, and cost of the humidity sensors can be further improved in the future, revealing potential for pharmacy, semiconductors, facility and equipment protection, and chemical and explosive storage.

## Figures and Tables

**Figure 1 sensors-20-05469-f001:**
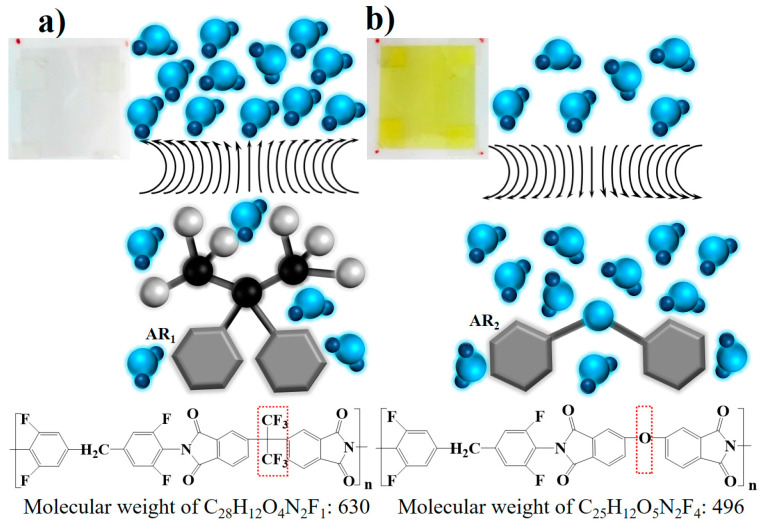
Chemical modification of the fluorinated polyimide (**a**) and non-fluorinated polyimide (**b**).

**Figure 2 sensors-20-05469-f002:**
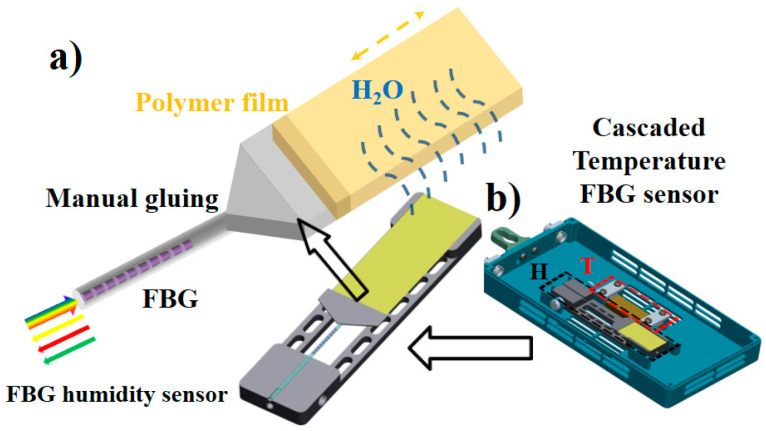
Sensing mechanism and separate film-grating structure (**a**)with another cascaded FBG temperature sensor to exclude the thermal crosstalk (**b**) in fiber Bragg grating (FBG) humidity sensor.

**Figure 3 sensors-20-05469-f003:**
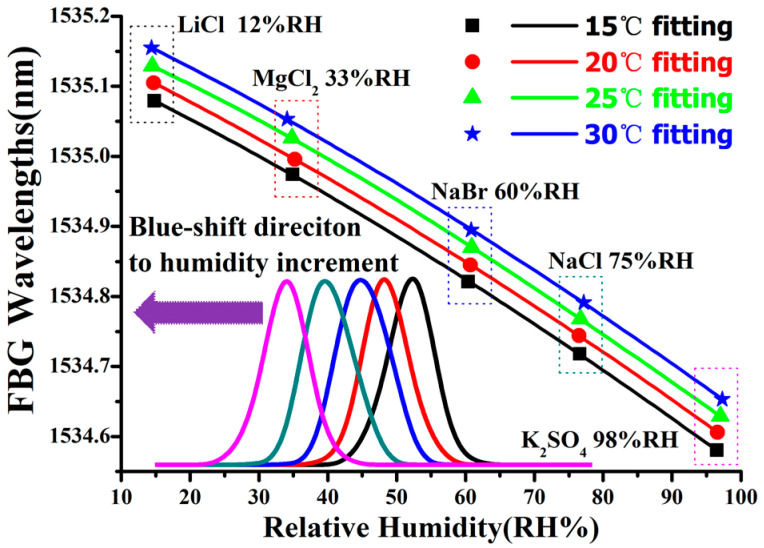
Humidity-dependent wavelength shifts with different saturated solutions and temperatures.

**Figure 4 sensors-20-05469-f004:**
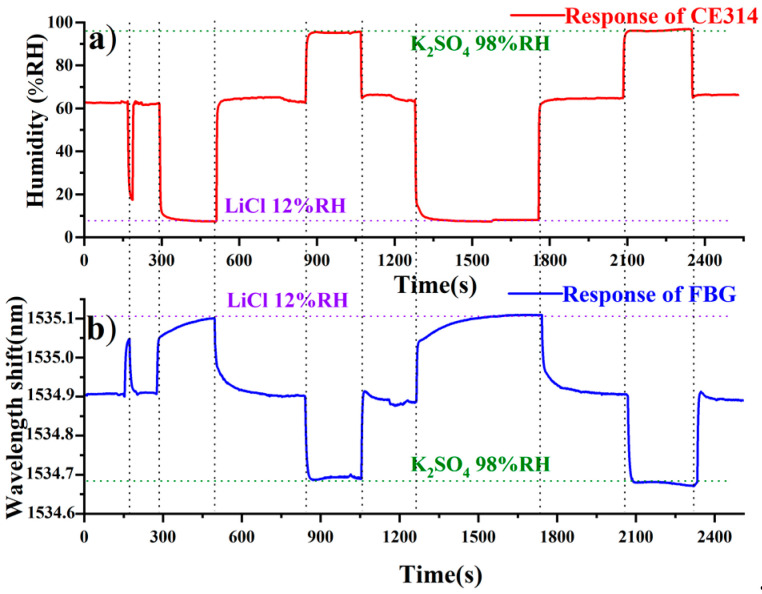
Responses to different humidity enclosures at 12% RH (LiCl), 62% RH (in the lab), and 98% RH (K_2_SO_4_) recorded by the CE314 sensor (**a**) and the FBG humidity sensor (**b**).

**Figure 5 sensors-20-05469-f005:**
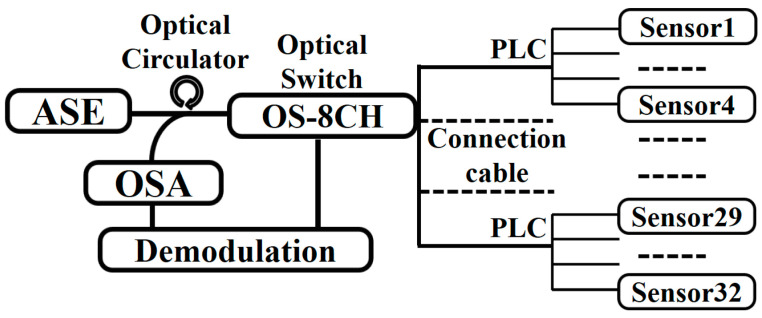
Brief illustration of the multiplexing sensing system with 8CH*4 sensors.

**Figure 6 sensors-20-05469-f006:**
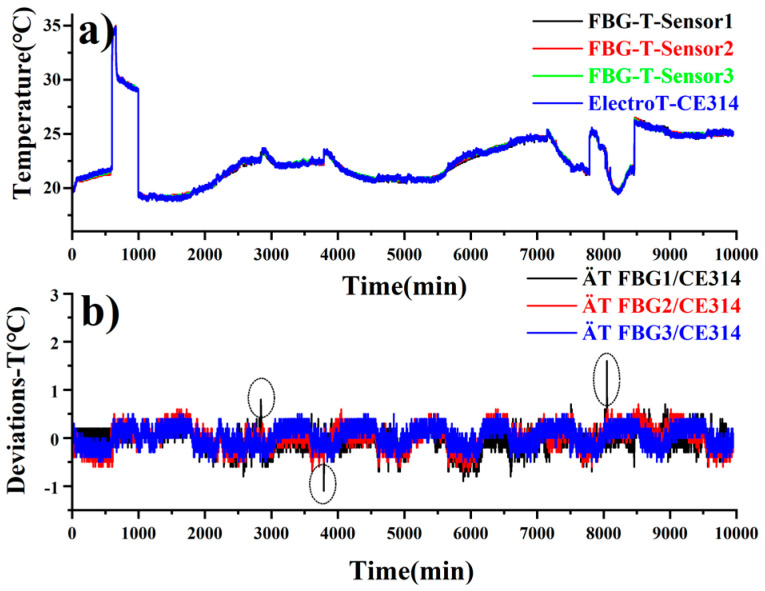
Temperature variations recorded by 3 FBG sensors and the CE314 sensor (**a**); temperature deviations between these 3 FBG sensors and the CE314 sensor (**b**).

**Figure 7 sensors-20-05469-f007:**
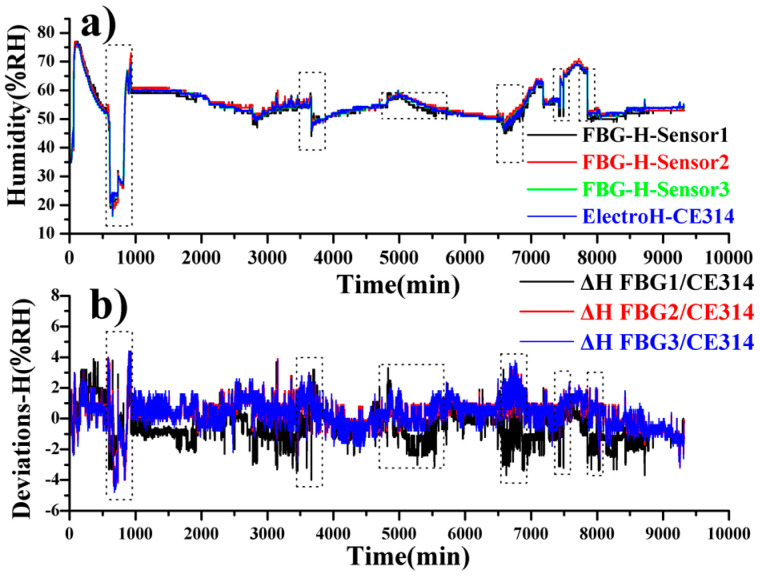
Humidity variations recorded by 3 FBG sensors and the CE314 sensor (**a**); humidity deviations between these 3 FBG sensors and CE314 sensor (**b**).

**Figure 8 sensors-20-05469-f008:**
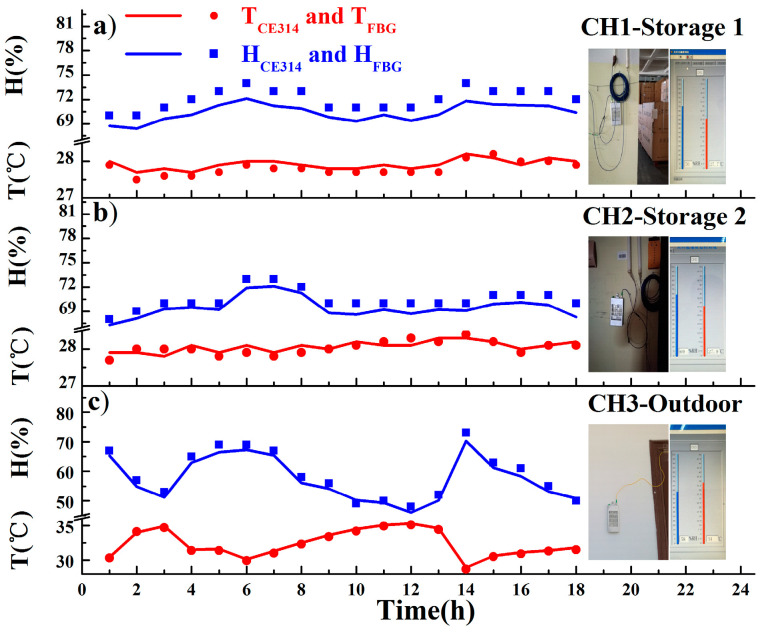
Temperature and humidity recorded by the FBG-1 and CE314 sensor at Channel1 in Storage 1 (**a**), by the FBG-2 and CE314 sensor at Channel 2 in Storage 2 (**b**), and by the FBG-3 and CE314 sensor at Channel 3 outdoors (**c**).

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
