# Peer review of "Fluorinated Polyimide-Film Based Temperature and Humidity Sensor Utilizing Fiber Bragg Grating"

_sensors, 2020, doi:10.3390/s20195469_

Round 1

Reviewer 1 Report

This work is devoted to the creation of a humidity sensor based on fluorinated polyimide. The research is inspiring, but there is no description of the chemical part of the process. It is necessary to add a diagram describing the chemical formula of the polyimide used and how it was formed and by what. An illustration showing Bragg diffraction when exposed to a sample is also missing. There are also a number comments:

1) line 34-35, links should be on each example, not at the end

2) line 37-38, links should be on each example, not at the end

3) line 55, novelty of the work should be explained in details (chemical formula of polyimide, difference of proposed polyimide from other polymers and difference of proposed concept from other)

4) Section materials and methods should describe polymer ctructure, polymer characteristics if it was bought or synthesis if polymer was synthesized by the group

5) Results and discussions should be improved by adding paragraph which explains the chemistry and physics of sensor and polymer characteriztion. How it was fluorinated, how the degree of fluorination was estimated, what is appearance of polyimide film?

Reviewer 2 Report

The authors present the construction of a multicomponent optical humidity sensor using a fluorinated polyimide film. Fluorination of the film enhances its optical transmission but lowers its moisture adsorption capacity, so the film is coupled to a fiber Bragg grating for enhanced sensitivity. The device operates by transferring moisture-induced strain from the film to the fiber Bragg grating resulting in a humidity-dependent wavelength shift.

The authors make a clear case for the value of the device in situations where conductivity or capacitance-based sensors could be hazardous or sensitive to interference. All familiar colorimetric humidity sensors are, as the authors point out, based on materials such as CoCl2 that either require regeneration or exhibit at best a very slow response to changes in humidity.

One issue with the study is that the light source, the ASE, is to my knowledge a laser that must be pumped with energy to produce light. It is not clear if this presents a disadvantage for the safety aspects that are specifically presented as advantages of this approach.

From the description of the spectral analyzing module, I also get the impression that the wavelength shift is detected using some sort of computer analysis. Again, in contrast to typical colorimetric humidity detectors, that means that this device only works under constant power. These limits warrant some discussion.

The method for correcting for thermal expansion is not entirely clear. On page 3 it appears that a second FBG is used as a temperature sensor to correct for the response to temperature in real-time; if so, some details are needed to explain how this second sensor can be included in the same optical path as the first sensor and kept at constant humidity. Alternatively, it appears from the discussion on page 4 that the device is simply calibrated across a humidity range for every temperature, and the calibrations are stored in the device. In that case, then again this is an electronic sensor that may share some of the disadvantages of sensors that the authors are looking to replace.

Reviewer 3 Report

The article is interesting and clearly written. However, it could be improved by taking into account the following points.

1. The Authors do not show how their solution differs from similar FBG solutions.

2. There is no comparison to solutions with FBG coating.

3. Has this functional material already been used with FBG?

4. It would be nice to give some error values.

5. No references to literature in chapter 2 and the following. Was this content entirely proposed by the Authors?

6. The conclusions could be extended a bit.

Round 2

Reviewer 1 Report

The sourse of used polymer should be mentioned in section materials and methods. In proposed version is unclear was the polymer synthesized by authors or it is commercially avaliable? In first case synthesis procedure should be describes, in second case name of supplier should be mentioned. In both cases molecular weight of used polymer shoud be mentioned
